# Generating intense electric fields in 2D materials by dual ionic gating

Benjamin I. Weintrub [ORCID][1], Yu-Ling Hsieh[1,2], Sviatoslav Kovalchuk[1], Jan N. Kirchhof [ORCID][1], Kyrylo Greben[1] & Kirill I. Bolotin [ORCID][1] ✉

The application of an electric field through two-dimensional materials (2DMs) modifies their properties. For example, a bandgap opens in semimetallic bilayer graphene while the bandgap shrinks in few-layer 2D semiconductors. The maximum electric field strength achievable in conventional devices is limited to ≤0.3 V/nm by the dielectric breakdown of gate dielectrics. Here, we overcome this limit by suspending a 2DM between two volumes of ionic liquid (IL) with independently controlled potentials. The potential difference between the ILs falls across an ultrathin layer consisting of the 2DM and the electrical double layers above and below it, producing an intense electric field larger than 4 V/nm. This field is strong enough to close the bandgap of few-layer $WSe_2$, thereby driving a semiconductor-to-metal transition. The ability to apply fields an order of magnitude higher than what is possible in dielectric-gated devices grants access to previously-inaccessible phenomena occurring in intense electric fields.

Electric fields are widely used to control material properties and to explore diverse physical phenomena. The first group of phenomena appears due to changes of the carrier density induced in a material by the field at its surface, typically explored using field-effect transistors (FETs)[1,2]. Electric fields also cause a second, qualitatively different, group of effects when the field penetrates through the material's bulk. In this case, the presence of a field inside the material breaks symmetries[3–7], bends the band structure along the direction of the field[3,5–7], and modifies the energetics of excitons with a dipole moment parallel to the field[3–6,8,9]. In conventional FETs, the induced carriers at the material's surface almost completely screen the field in the bulk of the material. Therefore, a dual gate FET with a pair of gate electrodes above and below the material under study is used to explore the effects of an external electric field penetrating a material. In this configuration, the field strength is controlled by the potential difference between the bottom and top gates, while the Fermi level is determined by their sum[5,7–9]. However, an important limitation for studying large electric fields in conventional solid-state FETs is the breakdown of gate dielectrics happening at around 0.3 V/nm[10–16] (somewhat larger dielectric strengths, ~1 V/nm, are measured using local probe techniques[11,12]).

The limitation on the maximum achievable carrier density has been overcome during the last decade via ionic gating, which combines condensed matter physics with electrochemistry[17,18]. In that technique, an ionic compound such as ionic liquid (IL), a molten salt, is placed over a material under study[17–20]. A potential applied between the gate electrode inside the liquid and the sample falls predominantly over an atomically thick (≤1 nm) electric double layer (EDL) at the IL/sample interface, modeled as a capacitor with an exceptionally large geometric areal capacitance (~10 μF/cm²)[17,19,21–27]. The resulting electric field inside the EDL induces a carrier density inside the material[27,28]. Critically, the field generated here is not limited by the dielectric breakdown of gate dielectrics, which limit the performance of conventional solid-state FETs. Instead, the only significant limitation is electrochemical modification of the material or electrodes, which occurs when the potential drop across a particular interface is too large (outside the electrochemical window)[27,29]. Ionic gating enabled previously inaccessible carrier densities larger than $10^{14}\,cm^{-2}$ to be reached[19,28,30,31]. The interactions between electrons at these carrier densities result in structural phase transitions[32] and electronic phases such as exotic superconductivity[30,31] and gate-controlled ferromagnetism[33,34]. These effects are especially pronounced in

---

[1]Department of Physics, Freie Universität Berlin, Berlin, Germany. [2]Department of Mechanical Engineering, National Central University, Taoyuan City, Taiwan. ✉e-mail: bolotin@zedat.fu-berlin.de

2DMs such as graphene, transition metal dichalcogenides (TMDCs), or phosphorene, where the carriers are spatially confined to one or few atomic layers[18].

Despite the progress in using ionic gating to induce high carrier densities, dual ionic gating has not been used to generate intense external electric fields inside the bulk of materials. Although single-gated suspended 2DMs[35,36] as well as hybrid dielectric/ion approaches to dual gating[37–39] have been used, no ionic counterpart to dual gate FETs has been demonstrated. Because of that, a wide range of phenomena predicted to emerge at fields stronger than $F_\perp \sim 1$ V/nm remains inaccessible in solid-state devices. For example, for fields near $F_\perp \sim 2$–3 V/nm, the interlayer bandgap of bilayer (2L) TMDCs is expected to decrease to zero[40,41]. In this situation, interlayer excitons should start forming at zero energy costs and a transition into a different state of matter, an interlayer excitonic insulator[42], may occur. Other predicted yet unobserved phenomena at intense fields include an insulator to topological insulator transition in phosphorene ($F_\perp > 3$ V/nm)[43], topological insulator to semimetal to normal insulator transition in 1T' TMDCs ($F_\perp > 2$ V/nm)[44], structural change in chirality for monolayer Te ($F_\perp > 7$ V/nm)[45], giant valley polarization ~65 meV in WSe$_2$/CrSnSe$_3$ heterostructures ($F_\perp \sim 6$ V/nm)[46], field-dependent magnon dispersion in 1L and 2L Fe ($F_\perp > 2$ V/nm)[47], and interlayer exciton condensates with high oscillator strength[48].

Here, we develop a double-sided ionic gating approach to generate ultrastrong electric fields. The approach can be viewed as a counterpart to conventional dual-gated FETs which is not limited by the breakdown of gate dielectrics. To generate the field inside a 2DM, we apply a potential difference between the two ILs above and below the 2DM, thereby generating a different type of EDL consisting of two ILs separated by an ultrathin membrane. We use a combination of electrochemical and electrical transport measurements to observe an electric field of more than 4 V/nm inside 2DMs, over 4 times larger than the biggest fields reported for conventional solid-state FET technologies and an order of magnitude larger than hBN-encapsulated devices.

## Results

### Device concept

At the core of our approach to generate and measure large perpendicular electric fields is an electrically-contacted few-layer 2DM suspended between two volumes of IL (Fig. 1a). The potential difference between the top and bottom ILs, $\Delta V^{ref}$, is controlled by separate top and bottom gate electrodes in contact with their respective ILs. This potential

difference falls across an ultrathin capacitor, thickness $d_\perp$, consisting of the 2DM (thickness $Nd_{int}$ for $N$-layer TMDCs with interlayer spacing[49] $d_{int} \approx 0.6$ nm) and two EDLs (average thickness[21–26] $d_{EDL} \approx 0.5$ nm), one above and one below the 2DM. We calculate the field inside a charge-neutral 2DM as $F_\perp = \Delta V^{ref}/d_\perp$, where $d_\perp = Nd_{int} + 2(\varepsilon_{2DM}/\varepsilon_{EDL})d_{EDL}$, $\varepsilon_{2DM}$ is the dielectric constant of the 2DM, and $\varepsilon_{EDL}$ is the dielectric constant of the EDL (see Supplementary Note 1 for the derivation). Approximating $\Delta V^{ref} \approx 5$ V (corresponding to the potential of top and bottom ILs at ±2.5 V, near the limits of the electrochemical window of our IL), $\varepsilon_{2DM} \approx 7.5$ for WSe$_2$[50], and $\varepsilon_{EDL} \approx 15$ for DEME-TFSI[35], we can estimate a maximum achievable field strength of at least 4.5 V/nm for our 1 L devices, at least four times larger than what is possible for the best dielectric-based devices and over an order of magnitude larger than fields achievable in hBN-encapsulated devices. The maximum achievable field reduces with $d_\perp$.

We determine the field strength using a specific 2DM, few-layer WSe$_2$, as a field sensor. We use electrical transport measurements to determine the bandgap of the 2DM[19,27], for which multilayers are known to exhibit a strong dependence on perpendicular electric fields[9,40,41]. When no field is present, the band structure of few-layer WSe$_2$ can be approximated as energy-degenerate conduction and valence bands, one set of bands per layer. An external perpendicular electric field breaks the inversion symmetry of the 2DM, thereby inducing a maximum energy difference of $e(N-1)d_{int}F_\perp$ between its outer layers, where $e$ is the elementary charge and $(N-1)d_{int}$ is the distance between the centers of the outer layers of the $N$-layer 2DM. Correspondingly, the bandgap of multilayer WSe$_2$ becomes spatially indirect, occurring between one layer's conduction band and another layer's valence band (Fig. 1b). We can estimate that the bandgap reduces from the field-free value, $E_N^0$, down to $E_N = E_N^0 - e(N-1)d_{int}F_\perp$ (Supplementary Note 1). Therefore, by extracting $E_N$ from electrical transport measurements, we can directly determine the field strength. The analysis above neglects free carrier screening, and it therefore only applies when the Fermi level of the multilayer is positioned within its bandgap. Nevertheless, the simulations of dual-gated multilayers accounting for screening agree with this simple model (Supplementary Fig. 4). Furthermore, photoluminescence data[49] and detailed DFT calculations[51] for bilayer WSe$_2$ predict a linear dependence of interlayer bandgap on the interlayer field with $e^{-1}dE_{2L}/dF_\perp = d_{int} \approx 0.6 \pm 0.1$ nm, very close to the interlayer separation in 2L WSe$_2$.

### Bandgap determination in dual IL-gated devices

We fabricate the dual IL-gated FET (Fig. 2) using standard 2DM processing techniques and measure it at room temperature in vacuum (see "Methods" for details). We measure a map of $I_{ds}$ vs. ($V_b$, $V_t$) for a bilayer (2L) WSe$_2$ device, sample #1 (Fig. 3a). Assuming equal coupling of top and bottom ILs to the material, the Fermi level of the system is controlled by $V_g = V_b + V_t$, whereas the field across the 2DM depends on $\Delta V = V_b - V_t$ (Supplementary Note 1). In constant-$\Delta V$ traces, we observe ambipolar transport with a region of negligibly low current (Fig. 3b). This region corresponds to the Fermi level ($E_F$) inside the bandgap of the material, while the areas of conductance to the left or right of that region correspond to $E_F$ within the valence or conduction band, respectively[5,9,19,33,52]. When the field is increased ($\Delta V > 0$ V), the region of zero current in Fig. 3b shrinks. That behavior is particularly clear in Fig. 3a where the region of zero conductivity (black region) along the $V_g$ direction gradually reduces with increasing $\Delta V$. Quantitatively, the bandgap is calculated from transport measurements as $E_{2L} = \frac{1}{2}e\alpha(V_e - V_h)$, where $V_e$ and $V_h$ are conduction and valence band threshold voltages, respectively (which correspond to the Fermi level at the conduction and valence band extrema)[27], and $\alpha \approx 0.77$ is the gating efficiency for this device (Supplementary Note 1). The bandgap determined in this manner decreases with $\Delta V$ and is linearly dependent on it (Fig. 3c). From the linear fit to this graph, we find

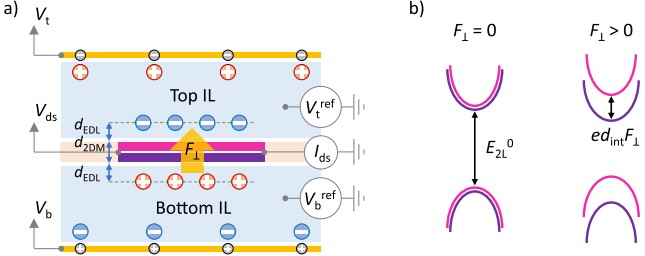

**Fig. 1 | Dual ionic liquid gating. a** Concept of a dual ionic liquid-gated bilayer 2DM device. The potentials of the top ($V_t^{ref}$) and bottom ($V_b^{ref}$) ionic liquids (ILs) are independently controlled by the top ($V_t$) and bottom ($V_b$) gate voltages. The potential difference between top and bottom drops over a ~2 nm thick layer consisting of two electrical double layers (thickness $d_{EDL}$) and the bilayer (thickness $d_{2DM}$), thereby generating an intense electric field ($F_\perp$) through the bilayer. The drain-source voltage ($V_{ds}$) controls a drain-source current ($I_{ds}$) through the 2DM. **b** A band structure sketch of a bilayer 2DM at zero and non-zero electric field. When no field is applied through the bilayer, the bandgap is $E_{2L}^0$. The field breaks the degeneracy between the energy bands corresponding to opposite layers and reduces the overall bandgap by $ed_{int}F_\perp$, where $d_{int}$ is the 2DM interlayer distance and $e$ is the elementary charge.

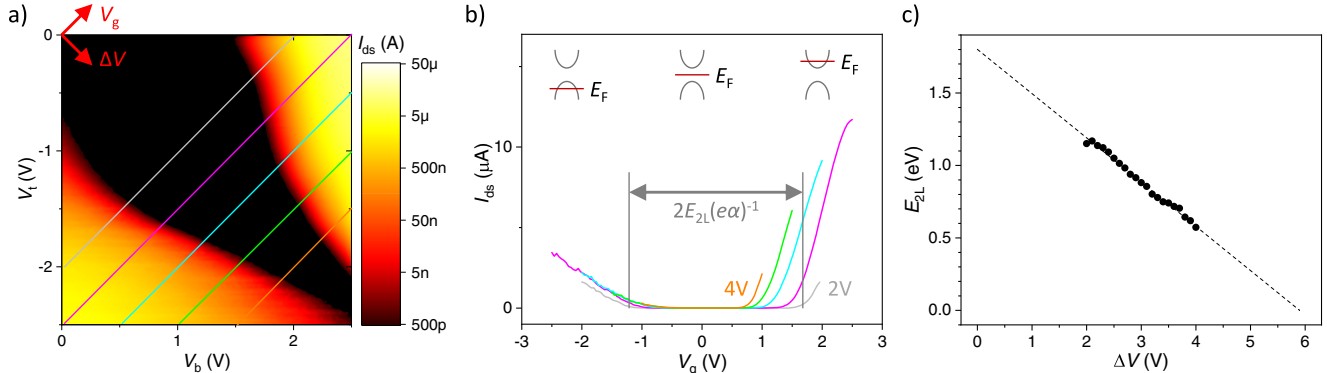

**Fig. 2 | Device and measurement overview. a** Side-view cartoon of the device and measurement scheme. The 2D material (2DM) is suspended over a hole in a silicon nitride (SiN) membrane grown on a silicon (Si) substrate. Gold (Au) electrodes are used to electrically contact the device, and cross-linked PMMA resist covers the drain/source electrodes. All voltages are defined as in Fig. 1. **b** Microscope (5x) image of sample #1 before applying ionic liquids. Drain and source electrodes are covered with cross-linked PMMA. Scale bar is 1 mm. Inset: 100x image of the 2DM deposited onto a square hole (~4 μm × 4 μm) in SiN. Everything in the inset other than the area of the 2DM over the hole is covered by cross-linked PMMA. Scale bar is 4 μm. **c** Photograph of sample #1 before measurement. Note that the top and bottom ionic liquids are not in contact.

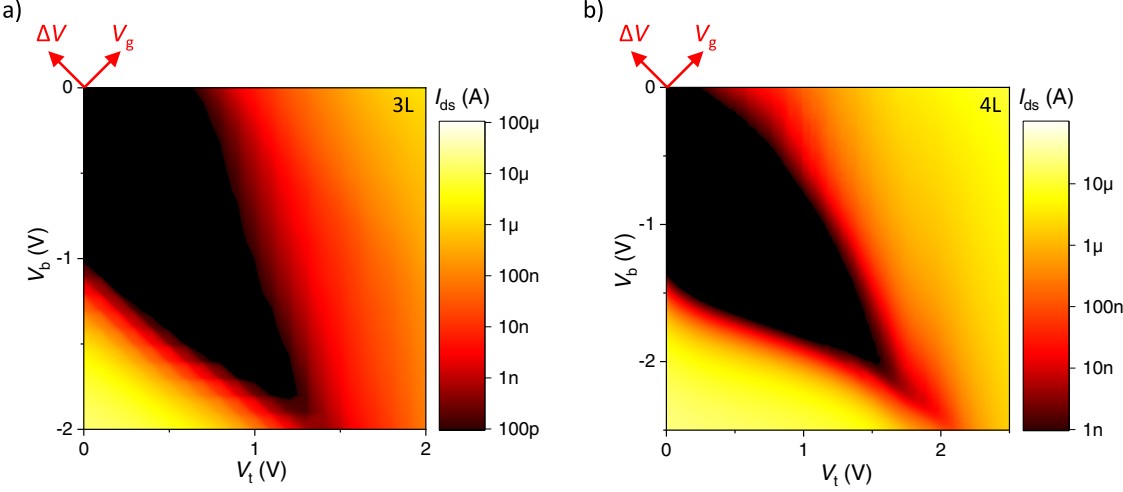

**Fig. 3 | Transport measurement of dual ionically-gated bilayer WSe₂. a** Map of $I_{ds}$ vs. $(V_b, V_t)$ for bilayer WSe₂ (sample #1). The axes of the coordinate system defined by $V_g \equiv V_b + V_t$ and $\Delta V \equiv V_b - V_t$, are indicated. The diagonal lines indicate constant $\Delta V$ from 2 V to 4 V. **b** Line scans corresponding to the slices of $\Delta V$ referenced in the map. The bandgap, $E_{2L}$, at each $\Delta V$ is obtained from the difference between threshold voltages, indicated for the gray curve ($\alpha$ is the gating efficiency and $e$ is the elementary charge). The bandgap shrinks with increasing $\Delta V$. Cartoons indicate the position of the Fermi level at different $V_g$. **c** Measured values of $E_{2L}$ vs. $\Delta V$ extracted from the line scans as well as a linear fit (dashed line).

**Fig. 4 | Transport in trilayer and quadlayer WSe₂.** Map of $I_{ds}$ vs. $(V_b, V_t)$ for **a** 3L WSe₂ (sample #2) and **b** 4L WSe₂ (sample #3). As $|\Delta V|$ increases from 0 V, the bandgap region (black) shrinks, and it eventually closes at a critical value of $F_\perp$ around 1.4 V/nm and 0.9 V/nm for the 3L and 4L device, respectively.

that the bandgap reduces to zero at $\Delta V_{close} \approx 5.9$ V. Using $\Delta V_{close} = E_{2L}^0 d_\perp/(e\alpha d_{int})$, we can experimentally determine the thickness of $d_\perp$, which is related to the EDL thickness, $d_{EDL}$. We obtain $d_{EDL} \approx 0.32 \pm 0.05$ nm, smaller than the value of ~0.5 nm quoted in the literature[21–26]. This discrepancy could stem from the redistribution of charges inside the ions or signal the limitations of our model.

Next, we note that in our geometry the same $\Delta V$ should produce a larger bandgap reduction in a few-layer 2DM compared to a bilayer. This is simply the result of a larger fraction of $\Delta V^{ref}$ falling across the 2DM compared to the EDLs as $N$ increases. A stronger response of few-layers to $\Delta V^{ref}$ allows reaching a field-driven semiconductor-to-metal transition. To observe it, we measure a 3L and 4L WSe₂ device (samples #2 and #3, Fig. 4). In contrast to the 2L device, we observe bandgap

a)

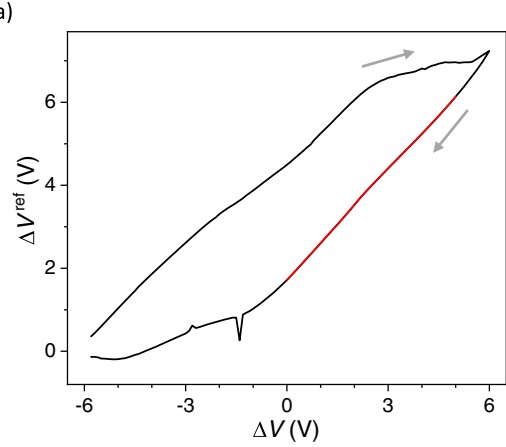

b)

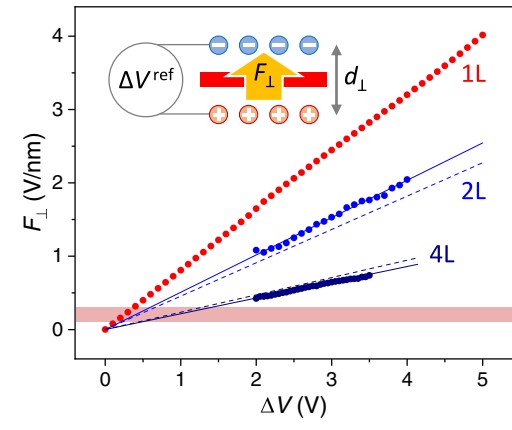

**Fig. 5 | Electric field through 1L, 2L, and 4L WSe₂. a** Reference voltage data from 1L WSe₂ (sample #4). The direction of the sweep is indicated by arrows, and the red portion of the curve is used to extract the field. **b** Perpendicular electric field through 1L WSe₂ extracted from reference voltage data (red data points) alongside the 2L and 4L WSe₂ field data extracted from transport maps (blue and dark blue dots, respectively) and fits to these data points (solid lines). Dashed lines for 2L and 4L devices represent the field values from estimated reference voltage, $\Delta V^{ref} = \alpha \Delta V$, for the respective devices. The red-shaded region from 0.1 to 0.3 V/nm shows the maximum field strength in conventional dielectric devices. The cartoon inset illustrates the relation between the potential difference of top/bottom ILs and the field across the device, $F_\perp = \Delta V^{ref}/d_\perp$.

closing at around $\Delta V \approx 4$ V for both devices, well within the electrochemical window. Bandgap closing is confirmed via measurements of the 'off' current vs. $\Delta V$ (Supplementary Fig. 9). When $\Delta V$ is increased further, electrons and holes are localized on opposite outer layers of the material (Supplementary Fig. 4). Exotic electronic states including excitonic insulators may arise in that situation[42,53].

### Quantifying field strength

Finally, we turn to the central result of this paper, the determination of field strength in our devices. For the bilayer device of Fig. 3, we determine the field as $F_\perp = (E_{2L}^0 - E_{2L})/(ed_{int})$ where we recall that $E_{2L}$ depends on $V_e - V_h$ (Fig. 5b, blue). The linear fit to these data points (solid blue line) indicates that the largest field achieved in that device at the maximum $\Delta V = 5$ V is $F_\perp \approx 2.5 \pm 0.3$ V/nm. The same procedure works for thicker devices (Fig. 5b dark blue shows the field measured in a 4L device) featuring stronger field dependence of the bandgap. The bandgap reduces to zero at $F_\perp \approx 1.4$ V/nm for the trilayer and ≈0.9 V/nm for the quadlayer. We note that screening effects dominant after closing of the bandgap prevent us from reliably measuring fields stronger than $F_\perp \approx 0.9$ V/nm in our 4L device. We expect that the smallest $d_\perp$, and hence the largest $F_\perp$, should be reached for monolayers. Unfortunately, the bandgap of monolayer WSe₂ does not depend on $F_\perp$ (Supplementary Fig. 3). Therefore, transport measurements cannot be used to determine the field. Instead, we rely on electrochemical measurements. For 1L WSe₂ (sample #4), we sweep a potential difference between bottom and top gate electrodes, $\Delta V = V_b - V_t$, while measuring the potentials of top and bottom ILs, $V_t^{ref}$ and $V_b^{ref}$, respectively, as measured by corresponding reference electrodes. The potential dropping across the 2DM and the two EDLs, $\Delta V^{ref} = V_b^{ref} - V_t^{ref}$, depends on $\Delta V$ linearly (Fig. 5a). The hysteresis seen in such data is commonly observed in ionic gating and stems from the delayed response of ions. We calculate the electric field strength across the monolayer as $F_\perp = \Delta V^{ref}/d_\perp$ (Fig. 5b, red; the field estimated using the same procedure for the 2L and 4L devices is shown as dashed lines). The field through 1L WSe₂ determined in this fashion reaches $4.0 \pm 0.4$ V/nm at $\Delta V = 5$ V (corresponding to the maximum $\Delta V$ used in transport measurements) using the literature value for the EDL thickness $d_{EDL} = 0.5 \pm 0.1$ nm. We consider this a conservative estimate, as the field is even larger using $d_{EDL}$ estimated from transport measurements. This measured field surpasses the dielectric strength of common gate dielectrics such as hBN[10,11], SiO₂[12], SiN[13,14], and HfO₂[15].

### Discussion

The largest field strength in our measurements, at least 4 V/nm, should already be sufficient to induce multiple predicted but still unobserved electronic and structural phases in various 2DMs[40,41,43,44]. The maximum field strength we report in 2L WSe₂ is limited by the electrochemical window of the IL we use. Conversely, the maximum field we report in 3L and 4L WSe₂ is limited by the material we use to measure the field strength. The bandgap of 3L and 4L WSe₂ closes at $F_\perp \approx 1.4$ V/nm and $F_\perp \approx 0.9$ V/nm, respectively, preventing measurements of higher field strengths using transport measurements. Even larger fields can generally be achieved by reducing $d_\perp$ (using thinner 2DMs and/or using ILs with smaller double-layer thickness), increasing $\Delta V^{ref}$ (using ILs with larger electrochemical windows), or using large-bandgap materials to measure the field. It is interesting to note that the fields extracted from our measurements are on the same scale or higher compared to the dielectric strength of 2DMs and common dielectrics. While the maximum field in dielectric-based devices is limited by the breakdown of both the gate dielectric as well as the material under study, in ionic gating approaches the limitation is due to electrochemistry at interfaces. Electric currents across the IL/2DM interface can only flow when the potential difference across that interface exceeds its electrochemical window. In addition, our dual IL-gated design should be compatible with a large variety of two-dimensional/other thin materials, ILs, and various ionic compounds (e.g., ionic gels and polymer electrolytes[17]). We finally note that our device uses an EDL consisting of IL/2DM/IL instead of the conventional IL/metal electrochemical system. This different type of EDL may allow studying the dielectric breakdown of materials as well as the exploration of other phenomena in intense electric fields.

Note: After submission, we became aware of a manuscript by Domaretskiy et al. which reports closely related experiments[54].

### Methods

#### Fabrication

A ~4 μm × 4 μm hole is fabricated in a Silicon nitride (SiN) membrane (20 nm SiN on Si purchased from Norcada) using focused ion beam lithography. Flakes of 2D materials (WSe₂ due to its relatively small bandgap[55–57], purchased from HQ Graphene) were mechanically exfoliated onto a thin piece of PDMS (~0.006 inches thick, X4 retention factor purchased from GelPak). The flakes were subsequently transferred over the hole in the SiN membrane at 60 °C using a homemade

transfer system. We fabricate drain, source, and top gate/reference electrodes using standard electron beam lithography (EBL) followed by the evaporation of 3 nm Cr and 70 nm Au. After liftoff, we confirm that the suspended region of the 2DM is not ruptured (Supplementary Fig. 7). Fabrication for sample #2 ends here.

As a final step for samples #1, #3, #4, and #5, we cover non-suspended regions of the 2DM and the drain/source electrodes with a layer of cross-linked poly(methyl methacrylate) (PMMA) EBL resist to prevent the interaction of these regions with the IL (Supplementary Fig. 6). Fabrication for samples #1, #3, and #5 ends here. To fabricate sample #4, we performed this cross-linking step in a 1 mm × 1 mm area around the 2DM immediately following the transfer, and then removed the excess PMMA in acetone.

## Measurements

Immediately before the measurement, a drop of IL (DEME-TFSI[20]) is placed on the back of the Si/SiN chip, which is subsequently placed onto a sapphire chip with gold electrodes serving as bottom gate/reference electrodes. A drop of IL is deposited on top of the device such that it is not in contact with the bottom liquid, and the entire assembly is loaded into an electrical probe station where it is measured at room temperature and low pressure, ~$10^{-5}$ mbar (Fig. 2c and Supplementary Fig. 8a). The gate voltages are swept at 10 mV/s except for the 3L data, which is swept at 100 mV/s. The data in Figs. 3 and 4 use top and bottom gate voltages controlled by separate sourcemeters along with a third sourcemeter to apply a drain/source potential, $V_{ds}$ (100 mV for 2L WSe$_2$ (sample #1) in Fig. 3, 50 mV for 3L WSe$_2$ (sample #2) in Fig. 4, 100 mV for 4 L WSe$_2$ (sample #3) in Fig. 4, and 100 mV for 1L WSe$_2$ (sample #5) in Supplementary Fig. 3). The size limitation of a suspended device limits electrical measurements including, for instance, multiterminal Hall measurements.

The $\Delta V^{ref}$ vs. $\Delta V$ data in Fig. 5a is collected by applying a potential difference, $\Delta V$, directly between top and bottom gate electrodes and measuring the reference voltage of each ionic liquid (IL) with the 2DM electrically floating. Reference voltages are measured by sourcing exactly 0 A to the reference electrodes while measuring their potential (Keithley 2450). In addition, we try to minimize the area of the bottom reference electrode relative to the gate electrode in order to reduce effects from the reference electrode.

## Data availability

The data underlying Figs. 3–5 of the Main Text are provided as Source data files. Any additional data which support this study are available from the authors upon request. Source data are provided with this paper.

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

## Acknowledgements

We gratefully acknowledge Dr. Christian E. Halbig for help with optical experiments. This work was supported by the Deutsche Forschungsgemeinschaft (DFG)—Projektnummer 182087777—SFB 951, Collaborative Research Center TRR 227, and ERC Starting Grant No. 639739.

## Author contributions

B.I.W., Y.H., J.N.K., and K.G. fabricated the samples. B.I.W. and Y.H. conducted the transport measurements. B.I.W., S.K., and K.G. conducted the optical measurements. B.I.W. and S.K. developed the simulations. B.I.W. analyzed the data. K.I.B. conceived the approach and supervised the project. All authors contributed to writing the manuscript.

## Funding

## Competing interests

The authors declare no competing interests.
