## [Peer Review File · Nature Communications]

Generating intense electric fields in 2D materials by dual ionic gatingEditorial Note: This manuscript has been previously reviewed at another journal that is not operating a transparent peer review scheme. This document only contains reviewer comments and rebuttal letters for versions considered at *Nature Communications*.

Response to reviewers

We thank the reviewers for careful reading of our manuscript. In response to comments from the reviewers, we implemented the following changes. We measured a new bilayer WSe₂ device, which provided much cleaner data (new Fig. 3a) consistent with our theoretical model (new Fig. S4b). We improved the accuracy of our analysis and obtained more reliable estimates for electric field strength. Finally, since the part of our manuscript reporting transport measurements in the regime of a closed bandgap at ultrastrong fields was most concerning to the reviewers, we decided to exclude this extra data and re-focus the manuscript. We now focus our discussion on the development and applications of a new technique to generate intense electric fields penetrating through the bulk of a material. To strengthen our story, we added new data and removed the data which was not convincing to the reviewers. As a result, we completely rewrote the second part of the manuscript.

We hope that all of these changes strengthen and clarify our narrative, while answering the concerns of the reviewers.

Reviewer #1:

The manuscript entitled “Dual ionic gate transistor: a device to induce extreme electric fields in 2D materials” reported gating a bilayer suspended TMD device from both sides with ionic gating. The device configuration is very interesting and challenging. I think the paper is worth being reported in *Nature Nanotech* if we can confirm the scenario with more controlled experiments and analyses.

The missing points are the follows which should be specifically addressed in the revision.

1. This is not the first paper reported for preparing a suspended bilayer TMD and gets gated from both sides. The previous work (*Nature Nanotechnology* 14, 1123 (2019)) of gating suspended bilayer should be cited.

We now cite this paper in the introduction section in the paragraph where we mention hybrid approaches as well as another earlier paper where few-layer TMDCs were suspended and single-gated with ionic liquid (*Nano Lett.* 15, 8, 5284, 2015) in line 52 of the revised manuscript.

We note that the physics explored in these works is very different from that in our manuscript, because there is only a single gate in these papers whereas our device geometry utilizes two gates enabling us to study the effects of a homogeneous external electric field penetrating the bulk of the 2D material.

2. The technical details are not trivial here. The author mentioned about separated gating of two surfaces. It is not very obvious that two droplets are isolated. Connecting them will mediate the gate response. This point needs further clarification.

We confirm the separation between top and bottom reservoirs of ionic liquid in the following ways:

1. **Imaging.** We image (optically and via SEM) the SiN membrane separating the top and the bottom reservoirs before and after measurement. We confirm that the 2D material remains unruptured after measurement. As an example, below we show images before (left) and after (right) measurement of sample #1, our bilayer WSe₂ device (Fig. S1a).

2. **Transport data.** In the 2L and 3L devices, which were confirmed to be intact via imaging, we observe clear signatures of a perpendicular electric field such as the field-induced bandgap closing. We expect that if the top and bottom ionic liquid reservoirs were in contact with each other, the potential of the top and bottom reservoirs are the same and therefore no perpendicular field would be generated.
3. **Reference voltage data.** In reference voltage measurements where the top and bottom ionic liquid reservoirs are in contact with each other, we expect a near-zero difference between top and bottom ionic liquid potentials (reference voltages). To confirm this, we compared the reference voltage difference (ΔV^{ref}) vs. applied voltage difference (ΔV) for an intact sample (1L WSe₂, device #3) against a sample without a 2D material which causes top and bottom ionic liquids to touch through a hole in the SiN. As expected, we observe a much larger ΔV^{ref} for the intact device. In addition, we note orders-of-magnitude different leakage currents between ruptured and unruptured samples (non-zero current for an intact device is due to capacitive effects). Overall, electrochemical measurements allow us to quickly and effectively determine whether a sample is ruptured or not.

To clarify these points to the reader, we changed the optical images in Fig. 2b,c with images from a different device (2L WSe₂, sample #1), taken before measurement. In Fig. 2c, the top ionic liquid reservoir appears as a small blob in the center of the device, isolated from the bottom ionic liquid

reservoir by the suspended WSe₂. The optical and SEM images of the same device are shown in Fig. S1a, including images taken after measurement showing the suspended region to be intact and unruptured.

3. As the electric field from the ions is terminated by the free carriers induced in the layers, especially, by the free carriers in the layer adjacent to the accumulated ions, therefore, it is not clear, why 3V can be estimated through the whole sample taking the simple configuration as shown in the inset of Fig. 3. According to the structure of the Helmholtz double-layer, the voltage drop actually appears at the empty gap between ions and induced carriers for both surfaces (Fig. 1A). Therefore, it is not clear how large the field could be after being screened by the free carriers and what is the field strength/voltage drop in the middle of the flake. A more realistic estimation considering the effect of screening is crucial in this case.

All of the electric field analysis in the manuscript is done in the regime of zero carrier density, when the Fermi level is inside the material's bandgap. For example, in our transport data (Figs. 3 and 4), we determine the size of the field-dependent bandgap by sweeping the Fermi level from the bottom of the valence band to the top of the conduction band. In this regime where the Fermi level is inside the bandgap, there are no free carriers, and hence we believe all of the free carrier screening effects can be excluded.

To clarify this point to the reader, we completely rewrote the analysis part of the manuscript. We now discuss the bandgap determination separately from the extraction of electric field strength. Finally, we added the following clarifying sentence in lines 96 to 97 of the revised manuscript: *"The effects of free carrier screening are neglected in this analysis, because the Fermi level is inside the bandgap."*

4. Related to the point above, the authors have not considered the screening effect and layer dependence in gating the flake. Due to the strong ionic gating, the induced carriers can be very much localized in a single layer. As reported in a bilayer sample with double-side gating (Nature Nanotechnology 14, 1123 (2019)), ionic gating can form doping states that resemble parallel individual layers.

As also reported in many previous papers, the field-effect gating on layered semiconducting TMD show a clear layer-dependent carrier accumulation profile (PHYSICAL REVIEW RESEARCH 3, 023047 (2021) in three-layer and 5-layer devices (PRL 119,147002 (2017)). Especially, in the latter case, the electronic state in the top and bottom layer can be individually tuned without having significant interference with each other. In the simulating, this could be well understood because the strong field is almost terminated by a single layer of TMD, where ~90% of the induced carriers are accumulated after strong ionic gating by assuming a reasonable effective mass (Fig. 3, PRB 88, 054515 (2013)).

Therefore, a clear statement should be presented to clarify that the layer-dependent carrier distribution is not applicable or suppressed in the present case. Namely, the present result can exclude the possibility of forming two separated channels at the top and bottom of the flake, leaving the voltage drop equals zero through the flake.

To address this comment, we refocused the manuscript on the regime where the density of free carriers is zero (unlike the works cited by the reviewer). In this regime, screening effects

are absent. In addition to restructuring the manuscript as mentioned above, we changed the figures to clarify that our focus is on the regime when the Fermi level is inside the bandgap. Also, we removed the data acquired in the “metallic” regime, after the field is strong enough to close the bandgap. We agree with the reviewer that the physics in that regime where free carrier effects are important is very interesting and complex. We intend to focus on this conductive regime in a follow-up work independent from this manuscript due to its complexity and focus our current manuscript solely on generating a strong electric field penetrating the bulk of a material.

5. The possibility of having two parallel channels can be easily ruled out by measuring a thicker layered structure. There must be a thickness limit at which the bandgap tuning effect is suppressed. Namely, the field-effect is terminated by bottom and top surfaces by the induced carriers and the net E field penetrating the flake in the middle becomes zero. Experimentally, a very thick layer, say 20-30 layers, should be measured to confirm that the data shown in Fig. 5a is not observed in thick flakes.

Please see our response to the two points above. To recap: we refocused the manuscript to make it clear that we concentrate on the regime of zero carrier density. Although the effects of free carriers is an interesting and complex topic, it is beyond the scope of our current manuscript which is about generating strong electric fields penetrating through the bulk of a given material.

We also wanted to comment regarding the reviewer’s statement that “There must be a thickness limit at which the bandgap tuning effect is suppressed”. In the simplest possible model, consider the effect of an external uniform electric field, F , applied through a charge-neutral N -layered TMDC. The maximum field that we can apply through this material is limited by the electrochemical window of the electrolyte and is given by $F_{\max} = \Delta V_{\max} / (2d_{\text{EDL}} + Nd_{\text{int}})$, where $\Delta V_{\max} = 6$ V corresponds to the electrochemical window for our ionic liquid, $d_{\text{int}} \approx 0.6$ nm is the interlayer separation (and individual layer thickness), and $d_{\text{EDL}} \approx 0.5$ nm is the average distance between ions and their nearest TMDC layer. This field of course drops with N . The field will produce an energy shift between outmost layers of the TMDC given by $\Delta E = e(N - 1)d_{\text{int}}F_{\max} \approx e\Delta V_{\max}(N - 1) / (N + 2)$ under the approximation that $d_{\text{EDL}} \approx d_{\text{int}}$. This expression indicates that the energy shift *does not* go to zero when N goes to infinity. Of course, when the material becomes very thick, this naïve model breaks and bandstructures of the top and the bottom of the material must be considered separately. Nevertheless, we believe that for moderate thickness, the bandgap tuning effect should not be suppressed.

It is not clear what kind of highly conducting state (Fig. 5A, the upper part with large delta-V) was eventually formed after double-sided gating in a bilayer (with a hole-like carrier or electron-like carrier) whether we can view the system as a uniform system. I think an additional Hall effect measurement can clarify this issue. Basic properties of the highly conducting state can be easily checked through the measurement of the temperature dependence of resistivity. The R-T dependence can clearly show whether the gap is indeed reduced.

We agree that it would be very interesting to investigate this new state occurring after bandgap closing, especially with additional Hall measurements. Given our suspended device

geometry, the Hall measurements are very challenging and would require a different generation of samples with different device geometries. Based on the feedback of both reviewers we decided to focus this manuscript just on generation of ultrastrong electric fields in the regime of zero carrier density. We plan to study the regime of high carrier density in a follow-up work. As explained in our answers to the previous points, we rewrote the manuscript and rearranged the figures to make this focus clearer to the reader.

Reviewer #2:

This paper contains unusually little data - the five figures could almost be combined into one - and what data there is is not very pretty.

In the current manuscript, we measured more devices and reanalyzed the data to improve data quality and clarity. In particular, we believe that the new data from our bilayer WSe₂ device presented in the new Fig. 3a is much more noise-free and convincing compared to the bilayer data in the previous version of the manuscript. In the new version of the manuscript, we now also show a transport map from trilayer WSe₂ (new Fig. 4). Lastly, maps from mono, bi, and trilayer are shown alongside simulated results using our model in the new Fig. S4, which we believe facilitates easy comparison between data and theory. For convenience, below we show new Figs. 3 and 4.

Fig. 3

Fig. 4

It is being sold as worthy of a top-flight journal based mostly on the claimed demonstration the highest electric field achieved in "any electronic device to date." I do not find the claimed record field particularly convincing based on the data and analysis, but much more importantly I do not see why it is so important, to whatever extent it is justified. The magnitude of the electric field at the sample surface is determined by the properties of the ion/electron double layer and there is nothing different in that regard from the many previous works using ionic gating, so the electric field involved here is nothing special, whatever its magnitude. The difference in this work is that using separated liquid compartments, opposite-sign ions are present above and below the sample so a larger effective penetrating displacement field can be obtained. This seems reasonable and the findings support it: the contrast between the data for bilayer samples 3 and 4 and the monolayer sample 1 seems pretty convincing evidence that the transport gap effectively is closed due to this displacement field. I do think that this likely closing of the gap in a bilayer on its own is a noteworthy result, irrespective of claims about the field magnitude, and of interest to the ion-gating community. However, it is not of broader interest (for example, to someone like me) for a number of reasons. First, ions on the surface are really not the same thing as an electric field applied through a dielectric. To the extent that they correspond to an average electric field, that field is extremely disordered and this is very deleterious for all electronic properties. The gates constantly leak, i.e. there is ionic and redox current, whose effects are just compensated for and subtracted off more or less empirically. The voltages can only be changed very slowly and there is huge hysteresis (see, eg, figs 3 and S3) as well as steady degradation due to redox processes. It is hard if not impossible to cool because the ionic liquids are fragile at low T. (The claim in the intro that ionic gating has "revolutionized condensed matter physics" is very bold). All of these disadvantages relative to dielectrics mean that a result is only of broader interest if it achieve extraordinary/valuable new goals than cannot be reached otherwise and can clearly be exploited and developed for interesting purposes. Simply making bilayer WSe₂ conduct at all gate voltages does not count as such an achievement in my mind. Showing that its optical gap fully closes, or that low-temperature transport and magnetoresistance indicated transition to a gapless 2D semimetal, would be far more compelling. Unfortunately this is probably not possible because of all the problems mentioned above, but if the authors could add data demonstrating either of those things my view would be very different. If they could really study in detail the state where the gap closes and show it's an excitonic insulator that would be fantastic.

If we understand the comment of the reviewer correctly, they have two main concerns: 1) The generation of a record electric field at the strengths we demonstrate is not an important achievement in itself, and 2) our technique has multiple limitations (e.g. disorder effects, leakages, cannot cooldown, etc...) which prevents its broader use.

Regarding #1, we point out that there is a long list of predicted phenomena (including phenomena which occur at room temperature), that have not observed yet because of electric field limitations in solid-state devices. To cite just a few examples, new topological phases and structural phase transitions are expected in 2D materials such as TMDCs (*Sci. Rep.* 10, 6670, 2020), monolayer Te (*ACS Omega* 5, 29, 18213, 2020), and black phosphorus (*Nano Lett.* 15, 2, 1222, 2015). In another example, a large perpendicular field is expected to cause a giant

valley polarization, splitting the valley degeneracy by ~ 65 meV in $\text{WSe}_2/\text{CrSnSe}_3$ heterostructures at $F_{\perp} \sim 6$ V/nm (*npj 2D Mater. Appl.* 5, 10, 2021). We also believe that there will be many other unexpected phenomena in the regime of strong electric fields, as it always happens when the new technology enables scientific experiments in a new parameter regime. This, in our view, this more than justifies the utility of any approach capable of generating such large fields.

To clarify this point to the reader, we updated the relevant passage in the manuscript (lines 55 to 62): “For example, for fields near $F_{\perp} \sim 2 - 3$ V/nm, the interlayer bandgap of bilayer (2L) TMDCs is expected to decrease to zero. In this situation, interlayer excitons should start forming at zero energy costs and a transition into a new state of matter, an interlayer excitonic insulator, may occur. Other predicted yet unobserved phenomena at extreme fields include an insulator to topological insulator transition in phosphorene ($F_{\perp} > 3$ V/nm), topological insulator to semimetal to normal insulator transition in 1T' TMDCs ($F_{\perp} > 2$ V/nm), structural change in chirality for monolayer Te ($F_{\perp} > 7$ V/nm), and giant valley polarization ~ 65 meV in $\text{WSe}_2/\text{CrSnSe}_3$ heterostructures ($F_{\perp} \sim 6$ V/nm).”

Regarding #2, we respectfully disagree with the statements regarding the limitations of the impact of our technique. In our view, this criticism can be equally applied to every paper using ionic gating to generate novel electronic states. Dozens of such papers have been published in high-impact journals, for example: ion-gating-induced superconductor-insulator transition (*Nature* 472, 458, 2011), carrier density-induced superconductivity (*Science* 338, 6111, 1193, 2012), and ferromagnetism induction (*Nature* 563, 94, 2018). All of these papers, combined, gathered hundreds of thousands of citations. Multiple reviews on ionic gating and its use in electronic transport are available (*Nat. Rev. Phys.* 3, 508, 2021, *Adv. Mater.* 29, 1607054, 2017, and *Nat. Rev. Mater.* 5, 787, 2020). Nevertheless, we would like to provide a response defending both our work as well as the entire field of ionic gating.

First, the reviewer mentions that the electric field generated by ions in an ionic liquid “is extremely disordered and this is very deleterious for all electronic properties”. The field in our experiments should be as disordered as the carrier density in previous ionically-gated devices. These devices, in turn, have been used in hundreds of experiments, many of them requiring high field uniformity. For example, ionic gating has been used for bandgap measurements of 2D materials (*Nat. Rev. Phys.* 3, 508, 2021), induction of superconductivity (*Science* 338, 6111, 1193, 2012), induction of topological states (*Sci. Rep.* 10, 6670, 2020 and *Nano Lett.* 15, 2, 1222, 2015), and induction of ferromagnetism (*Nature* 563, 94, 2018 and *Nat. Comm.* 9, 1897, 2018). All of these experiments would not have been possible if ion-related disorder were dominant.

Furthermore, we carried out a quick characterization of disorder by measuring photoluminescence (PL) of ionically-gated SiO_2 -supported WS_2 . The width of the trion peak in the PL spectrum is commonly used as a rough disorder estimate. The evolution of the PL spectrum with gate voltage is shown below. The measured width of the peak, ~ 60 meV, is comparable with that of devices on a common dielectric such as SiO_2 (~ 50 meV, Fig. 2d from *Appl. Opt.* 55, 6251, 2016). This suggests that the level of disorder in fields produced from ionic liquid gating is at most similar to the level of disorder in the devices on SiO_2 gating. This

level of disorder was tolerable for hundreds of experiments on TMDs before the disorder was decreased in hBN-encapsulated devices.

The second concern is that “The gates constantly leak, i.e. there is ionic and redox current, whose effects are just compensated for and subtracted off more or less empirically”. Leakage currents are always less than ~ 1 nA, at least 1-2 orders of magnitude smaller than I_{ds} we use in bandgap analysis, and we use raw data without subtracting leakage when we extract the values of electric field. We simply take advantage of the fact that in the linear mode of transistor conduction, $I_{ds} \sim V_g - V_{threshold}$, so we form a linear extrapolation of the linear regime of the electron and hole regions of an I_{ds} vs. V_g curve to measure the values we need, which is typically done when I_{ds} is on the order of 100 nA to 1 μ A. This is the standard analysis used in many previous works (e.g. *Nat. Rev. Phys.* 3, 508, 2021).

To clarify this point to the reader, we updated the information about leakage current in the supplementary material: leakage current “in our devices is at most a few nA, at least one order of magnitude lower than the scale of I_{ds} used to obtain threshold voltages” in lines 131 to 132 of the SI.

The next concern is that “The voltages can only be changed very slowly and there is huge hysteresis (see, eg, figs 3 and S3) as well as steady degradation due to redox processes”. The measurement speed does not prevent us from acquiring clean data. The device is stable when the voltage is kept within the electrochemical window and swept with reasonable speeds (10 to 100 mV/s). The data referred to by the reviewer in the old manuscript was taken at ‘fast’ speeds of 100 mV/s, which may cause hysteresis, however this does not affect the strong fields we extract from the data. In our revised manuscript, the transport map for the bilayer device (Fig. 3a) was taken at 10 mV/s, and the map for the trilayer device (Fig. 4) was taken at 100mV/s. Both transport maps are clean, indicating that our sweep speeds do not noticeably affect our results. For convenience, below we show the transport maps for both the bilayer (Fig. 3a, left) and the trilayer (Fig. 4, right).

With regard to “steady degradation due to redox processes”, it is of course a concern. Nevertheless, the degradation is manageable within the electrochemical window of the electrolyte. Some of the devices are measured for weeks. The optical images taken before measurement indicate clean suspended WSe₂, and the SEM images taken afterwards indicate that the bilayer and trilayer sample remain unruptured (Fig. S1 of the supplementary info). We therefore believe that degradation is not significant in our devices.

The final concern is “It is hard if not impossible to cool because the ionic liquids are fragile at low T”. Here, we point out that many ionic gating experiments are done at low temperatures. Specifically, the ionic liquid is cooled through its freezing temperature while gate voltage is applied. It has been repeatedly shown that such procedure retains the high quality of the device as well as the applied field (*Nat. Nano.* 14, 1123, 2019, *Science* 338, 6111, 1193, 2012, and *Nature* 563, 94, 2018). In our experiments, which we provide the first demonstration of an approach, the need to cool down our devices did not arise. We think, however, that it will be possible with our approach as well given the success of freezing other ionically-gated 2D materials.

The claim in the intro that ionic gating has "revolutionized condensed matter physics" is very bold.

We removed this phrase.

All of these disadvantages relative to dielectrics mean that a result is only of broader interest if it achieve extraordinary/valuable new goals than cannot be reached otherwise and can clearly be exploited and developed for interesting purposes. Simply making bilayer WSe2 conduct at all gate voltages does not count as such an achievement in my mind.

To clarify: our work is not about closing the bandgap but rather about a new approach to generate the strongest static electric field ever recorded penetrating through the bulk of a device using a novel technology. We rewrote the manuscript to emphasize the point. We now stress that bilayers are simply used as a vehicle to quantify the field strength.

Although it's not my main problem with the paper, the claim of 3 V/nm of perpendicular electric field (bottom of p5) is certainly rough and likely optimistic. It seems to be based on just on an estimate of the effective width of the system and a straight line drawn through a

hysteresis loop, combined with DFT predictions that the WSe₂ gap closes at 3 V/nm. DFT is not very accurate for such things.

We would like to clarify that our measurement of field strength is based on two independent approaches each giving the expected results consistent with our model. The first approach is the direct measurement of the voltage drop across the ionic layers and 2D material, ΔV^{ref} , defined as the difference between top and bottom reference voltages. The field is then calculated as $\Delta V^{\text{ref}}/d_{\perp}$, where d_{\perp} is the combined thickness of the 2D material and the electrical double layers above and below it. The second approach is based on the field dependence of the bandgap of bilayer and trilayer WSe₂. Critically, the two independent approaches agree and give values consistent with what we expect from our model.

We re-structured the manuscript to better convey this point.

The reviewer says that “DFT is not very accurate for such things”. We agree. However, we can compare our results in the regime of small field to previous experimental work employing solid-state gating which uses the same field-dependent models for bilayers that we use but applied in the low-field regime (*Phys. Rev. Lett.* 123, 247402, 2019). Due to the fact that the models used in the literature match our models for the bilayer, and given that the model accurately predicts both the low-field behavior and the high-field behavior presented in our manuscript, we believe that our result is robust. Our extrapolation of the model to more than 2 layers accurately reflects what we see in the new 3L WSe₂ transport maps as well. In addition, the field we extract from reference voltage data is a direct application of the field-dependent part of our simulations, $F_{\perp} = \Delta V^{\text{ref}}/(2d_{\text{EDL}} + Nd_{\text{int}})$, where we directly measure ΔV^{ref} and we calculate d_{EDL} from our bilayer transport map. The values of electric field that we calculate from this data are similar to what we would expect for this device, supporting the assertion that our model is accurate.

To claim a record they need to demonstrate it accurately with small error bars, for comparison with dielectrics where the breakdown field is precisely measured.

We state that common dielectrics have a measured dielectric strength of ~0.5 to 1 V/nm in lines 29 to 32 of the revised manuscript, and our maximum measured field is larger than $\sim 3.5 \pm 0.2$ V/nm. We note that the field measured is much higher compared to the dielectric breakdown field due to the difference between ionic and electrostatic gating. We make this comparison in the abstract (lines 12 to 14)) and in the main text (67 to 70 and 157 to 163) of the revised manuscript.

Regarding error bars: to strengthen our claim, we took more measurements and fixed our analysis. The bilayer device in the revised manuscript shows much cleaner field dependence. We now estimate the field strength in a more robust way, by extracting threshold voltages using the linear regime of transistor conduction. We improved our error analysis for both transport and reference voltage experiments, along with the description of applicability limits (see lines 132 to 133 in the main text). Nevertheless, we point out that the electric field is measured in a regime it has not been studied before. Because of that, ultra-precise probes for an electric field of that strength are not available. We believe that our work will stimulate the development of such probes.

Incidentally, there's also a question of the how the polarizability of the WSe₂ screens the electric field of the ion layers. If, for instance, the effective out-of-plane dielectric constant of the bilayer WSe₂ is of order 10 (which I saw reported in one paper, but it seems rather uncertain) it will self-screen the external electric field down by up to a factor of 10. This may be a component in the DFT calculation, but it would seem to lead to needing a much larger external field than 3 V/nm to close the 1.9 eV gap. I confess I have not studied the literature, but the authors make no mention of the question which seems to me important.

Again, we would like to point out that our data on bandgap vs. field match with previously reported results reported using solid state gating (*Phys. Rev. Lett.* 123, 247402, 2019). Because of that, we believe that self-screening is accounted for correctly in the relevant calculations.

REVIEWER COMMENTS

Reviewer #1 (Remarks to the Author):

I've read the reply and revision of the manuscript entitled "Dual ionic gate transistor: a device to induce extreme electric fields in 2D materials". Compare with the 1st draft, I think several important improvements have been made. I tend to agree with the publication in Nature Communication.

In addition, I have the following summary and suggestions:

1) Details about the separated operation of two gates, including the image, transport data, and reference voltage data. Those are very helpful and important supporting information. I would suggest the author include their reply to me in the supporting information.

2) Clearer statement about their analysis has been added, which is in the regime of zero screen effect from free carriers. I am still in doubt whether this is entirely correct but it is at least more self-consistent.

But the question remains whether the carrier is indeed zero. The authors believe that they can deplete completely the carriers, but no clear evidence confirms that. Even if a metallic state is indeed formed, there will be a non-zero carrier, hence a nonzero screening effect.

3) The authors have correctly cited two previous reports of double-side ionic gating. Therefore, it is more complete for the references.

4) As the authors also agree that a definitive confirmation of the state eventually formed can be achieved by measuring the Hall effect. And if the gap is indeed closed, the metallic state should also show clear metallic temperature dependence. I think making a clear statement of the limitation of the present experiment is essential.

Reviewer #2 (Remarks to the Author):

The paper by Benjamin Weintrub and colleagues titled "Generating extreme electric fields in 2D materials by dual ionic gating" reports ionic gating of suspended TMDs from two sides. In other words, it substitutes solid dielectrics in traditional dual-gate geometry with electric double-layer transistor dual-gate geometry with much higher capacitance.

I believe that this is an important advancement in the field of ionic liquid gating and that the results (if some of them would be thoroughly justified during the revision process, see below) are general enough and interesting enough to the general audience to grant publication in Nature Communications.

However, before agreeing that this paper is ready for publication, I believe that the following most critical aspect of the work should be addressed:

1) The claim of closing the gap with electric field in trilayer WSe₂ reported in Figure 4 should be justified by either (i) temperature-dependent measurements, where the difference in temperature-dependent behavior at different doping levels and electric fields is discussed, or by (ii) optical measurements, where either PL or reflectance should be measured as a function of both gates. Room temperature transport alone cannot serve as a smoking gun evidence of gap closing in multilayers of TMDs. I do understand that temperature-dependent measurement (preferably four-terminal for careful accounting of contact resistance) in dual gate ionic liquid gated sample is extremely challenging. In case authors could perform PL or dR/R measurements instead and demonstrate clearly how optical response changes this would be a much stronger claim. Otherwise, the absence of a resistive region on the bottom of the 2D map in

Figure 4 could be explained trivially by the formation of two conductive layers on top and bottom surfaces but without modification of electronic structure. To prove that electronic structure is indeed changing it should be probed in an alternative way (either optics or T dependent transport measurements). The same question is valid for the interpretation of data on 2L WSe₂ in Figure 3. If this open question is addressed, the paper will become much stronger.

2) Related to previous point, but on a more abstract note, which will echo concerns of other reviewers – authors do say in the introduction to the paper that ultra-high electric fields could enable a lot of new experiments, but they do not show enough evidence for that. In case my point 1) is addressed then there would be more substantial evidence that it is indeed possible to do something exciting with those electric fields (change substantially or close the bandgap of TMDs).

Some other more technical questions:

3) To understand the methodology of reference potential measurement, have the authors tried to use a high impedance voltmeter (for example electrometer) instead of sourcing additional voltage to a reference electrode (which could act as a gate as well)? It might be worth providing a bit more details regarding the measurement of potential inside IL for general readers. It might be also worth explaining if biasing reference electrodes affects the distribution of potentials inside the IL.

4) SI Figure 3 contains a 2D map and linecuts for dual ionic liquid gating in monolayer WSe₂. What does the decrease of current to zero at negative gate voltages represent in linecuts? Is that degradation?

5) Have authors measured the capacitance of their ionic liquids?

Some technical comments to better represent the data:

6) Figures 1 and 2 could be merged. It is up to the authors of course, but they discuss two very related things in the form of schematics, and it could be beneficial for a potential reader to see the concept, physics of what is happening, and device schematic and image on the same figure. At the same time, each figure taken individually does not contain enough information in my opinion.

7) Supplementary Figure 1 – it might be worth increasing the size of device images, they are not readable, also please add scale bars.

8) Grey lines on 2D maps in Figure 3 and 4 should be described in the main text or Figure captions.

Reviewer #3 (Remarks to the Author):

The points raised in the previous round of review have been satisfactorily addressed. A significant advance of the present work over previous studies of tunable bandgaps in 2D materials using vertical electric fields -such as Nature 459, 820–823 (2009), Nano Lett. 15, 8000–8007 (2015), is that a much stronger electric field is achievable using dual ionic-liquid gates in this work.

However, I feel that the manuscript can be further improved by incorporating data the authors might have already taken.

1) As the bandgap is reduced by a vertical electric field, the off current is expected to increase. The authors should also replot Figure 3b on logarithmic scale to show how the off current varies with the vertical electric field in the supplementary information. Similar plots should also be made for the trilayer device.

2) Did the authors measure the PL of their bilayer devices as a function of ionic-liquid gate voltage? If so, did they observe the redshift of a peak corresponding to the interlayer transition, like figure 4 in Nano Lett. 15, 8000–8007 (2015)?

Response to reviewers

We would like to thank the editor and all reviewers for their careful reading of and comments about our manuscript. To respond to reviewers' concerns, we carried out several new experiments. First, we confirmed modification and closing of the bandgap in response to a perpendicular field via two new independent approaches: optical measurements and subthreshold current measurements. Both of these approaches confirm our model of field-dependent interlayer bandgap modification from two different perspectives; we thank the reviewers for suggesting these experiments. Second, we developed a simple model of bandgap modification in double-gated multilayer devices. The model explicitly includes screening by free carriers and outputs layer-resolved carrier densities. The model confirms our previous simplified approach (which ignored screening effects) to extract the bandgap and field strength from transport data; it introduces a small correction in the field extracted from reference voltage data by treating the ionic liquid/2DM interface more thoroughly. Finally, we fabricated and measured a new four-layer (4L) WSe₂ device which shows bandgap closing more cleanly. This new data and new understanding resulted in multiple modifications of the main text and supporting information. Two new persons assisting us with optical experiments were added to the authors list and acknowledgements, and one person who was previously just acknowledged was also added to the author list for significant work they did earlier in the project. We feel that all of these changes significantly strengthen and clarify our narrative while answering the concerns of the reviewers. We mark all changes in the main text and supplementary information in yellow highlight for reviewers' convenience.

We finally note that very recently, during the second review stage of the manuscript, a very related publication has been accepted to *Nature Nanotechnology* (DOI: <https://doi.org/10.1038/s41565-022-01183-4>). This work cites our arXiv submission (the arXiv version of the *Nat. Nano.* paper appeared simultaneously with ours). The results are similar in that both describe the approach to generate ultrastrong electric fields via double sided ionic gating and show that these fields are strong enough to close the bandgap of TMDCs. The two works complement each other since we both use dual ionic gating; however, our work uses two ionic liquids whereas their work uses a lithium-ion conductive glass substrate with ionic liquid on top. The techniques have complementary strengths and weaknesses. We think that the appearance of that work highlights the timeliness of our results and gives another perspective to them.

Reviewer #1:

I've read the reply and revision of the manuscript entitled "Dual ionic gate transistor: a device to induce extreme electric fields in 2D materials". Compare with the 1st draft, I think several important improvements have been made. I tend to agree with the publication in *Nature Communication*.

In addition, I have the following summary and suggestions:

1) Details about the separated operation of two gates, including the image, transport data, and reference voltage data. Those are very helpful and important supporting information. I would suggest the author include their reply to me in the supporting information.

We can interpret the request to demonstrate “the separated operation of two gates” in two different ways. One interpretation would be to demonstrate that top and bottom ionic liquids are not in contact and can be controlled separately. Here, our main arguments would be i) transport maps indicating reduction/closing of the interlayer bandgap (Figs. 3-4, such bandgap reduction should not occur if top and bottom liquids were in contact), and ii) comparison of reference voltage measurements between intact and ruptured devices (Fig. S7). We clearly see that when top and bottom ionic liquids are connected, their potentials equilibrate. In contrast, in unruptured devices we see a large difference in potentials between top and bottom liquids. There are also optical and electron microscopy images showing isolation of the liquid on top and on the bottom (Fig. 2c and S8a) as well as the image of the 2L device before and after measurement (Fig. S1), demonstrating that no macroscopic defects are created during the experiment.

Another possible interpretation is a request to show that top and bottom gates can be used independently to control the device. We agree that such information is useful, and we include it in a new figure in the supporting information, Fig. S8 (shown below as Fig. R1 for convenience). We acquired this data in a newly fabricated 4L WSe₂ device, where we show a) the assembled device with isolated top and bottom ionic liquid just before measurement, b) a scan of drain-source current as a function of bottom gate voltage with a fixed top gate of 0 V, and c) a scan of drain-source current as a function of top gate voltage with a fixed bottom gate of 0 V.

Fig. R1 | Individual gate operation. a) Optical photograph of a 4L WSe₂ device assembled in a probe station with visibly-isolated top and bottom ionic liquids, **b)** I_{ds} vs. V_b when $V_t = 0$ V, and **c)** I_{ds} vs. V_t when $V_b = 0$ V.

2) Clearer statement about their analysis has been added, which is in the regime of zero screen effect from free carriers. I am still in doubt whether this is entirely correct but it is at least more self-consistent.

But the question remains whether the carrier is indeed zero. The authors believe that they can deplete completely the carriers, but no clear evidence confirms that. Even if a metallic state is indeed formed, there will be a non-zero carrier, hence a nonzero screening effect.

We want to clarify that in the updated manuscript, we do not consider in detail the metallic regime. We only focus in detail on the region of near-zero current when the Fermi level is inside the bandgap of the material (the boundaries of that regime are used for field extraction). In that region, the conductivity of our sample is zero to within experimental uncertainty, and the density of free carriers is zero (to within measurement error), in accordance with the Boltzmann formula for conductivity. Using this formula, we can roughly estimate the maximum residual density of free carriers as $n = \sigma/\mu e$, where σ is the minimum conductivity corresponding to the minimum current we can measure, ~ 10 pA, and assuming a carrier mobility of $\mu \sim 100$ cm²/Vs. We get a vanishingly small carrier density of $\sim 10^7$ cm⁻². We consider this clear evidence that free carriers are depleted. We added the following clarification to the main text in lines 101 to 104:

“The analysis above neglects free carrier screening and is therefore only applies when the Fermi level of the multilayer is positioned within its bandgap. Nevertheless, the simulations of dual-gated multilayers accounting for screening agree with this simple model (Fig. S4).”

We interpret the reviewer’s mention of screening effects as the concern of screening due to either this small density of free carriers or due to carriers localized at the defects (which do not contribute to conductivity). To address this concern, we developed a new approach (based on model from *Phys. Rev. Lett.* 123, 117702, 2019) to calculate the response of multilayer TMDCs to electric fields which does include screening effects (our previous simulations were much simpler and that effect was ignored). The simulation solves for carrier densities inside each layer of the multilayer so that the total chemical potential is equilibrated throughout the device (details in the updated Supplementary material section S2 and Fig. S4).

Below in Fig. R2, we compare simulated carrier densities in: a) our previous model (in that model, we could only distinguish the regimes of zero and non-zero carrier densities), b) our new model with no localized carriers, and d) the new model including a reasonable density of localized states, $n_0 \approx 10^{12}$ cm⁻² (*Nano Lett.* 20, 2544, 2020). All simulations are for the same device, a double-gated bilayer TMDC, where the orange diamond shows the region where the Fermi level is inside the bandgap. In addition, the electric field between the TMD layers corresponding to b) and d) is plotted in c) and e) respectively.

Fig. R2 | 2L WSe₂ old vs. new simulations. For our 2L device, we show simulations using **a)** our old model indicating bandgap vs. conductive regions, and **b)** through **e)** are from our updated simulations (orange diamond shows bandgap region). **b)** and **c)** show the total number of carriers and the field between TMDC layers respectively for 0 localized carriers in the bandgap, and **d)** and **e)** show the same thing for 10^{12} cm^{-2} localized carriers in the bandgap. The contours in **c)** and **e)** show 1 V/nm intervals, starting from 0 V/nm (diagonal line) in both plots.

We see that the region of zero conductivity (which is used in the manuscript to extract the field strength and to detect bandgap closing) is virtually the same in both simulations. Moreover, a small remnant density of free carriers or localized states does not affect the shape of that region or the field inside it. Inside of the bandgap region, the field depends on ΔV linearly and is described by a simple formula given in the manuscript. In contrast, we clearly see that outside of the bandgap region the field drops as more carriers are brought into the material. This is the result of field screening by free carriers mentioned by the reviewers.

Our field extraction recipe relies on finding the point of bandgap closing, and therefore, we conclude that screening effects, while extremely important and interesting, do not affect the accuracy of field extraction. To clarify this to the reader, we completely re-wrote the section dealing with simulations in the supplementary material, and included a version of the figure shown here as new Fig. S4, and use the model everywhere in the text.

3) The authors have correctly cited two previous reports of double-side ionic gating. Therefore, it is more complete for the references.

4) As the authors also agree that a definitive confirmation of the state eventually formed can be achieved by measuring the Hall effect. And if the gap is indeed closed, the metallic state should also show clear metallic temperature dependence. I think making a clear statement of the limitation of the present experiment is essential.

We agree that Hall measurements are well-suited for the investigation of the metallic state, however we did not carry out these measurements for two reasons. First, in the updated version of the manuscript, the metallic state is not studied. Instead, the focus is on the insulating state and measurements of the electric field by studying the boundaries of that state. Hall measurements are not as useful to probe the insulating state. Second, it is challenging to carry out Hall measurements in our devices. Our dual gating technique puts boundaries on the device size and makes it difficult to define additional electrodes needed for Hall measurements. We added the following remark to the supplementary information in section S3 about transport and reference voltage measurements in lines 138 to 139:

“The size limitation of a suspended device limits electrical measurements including, for instance, multiterminal Hall measurements.”

The reviewer also suggests looking at the temperature dependence of electrical transport to confirm insulating and metallic character of resulting states. We followed this advice and carried out a version of that measurement. Since our measurement involves ionic liquids, the temperature cannot be changed directly in a wide range (the ionic liquid either freezes at low temperatures or degrades at high ones). However, we can do a very related measurement. Consider the “off” (subthreshold) current of our device, when the Fermi level is inside the bandgap, E_{gap} . That current is given by $I_{\text{OFF}} \sim n_{\text{intrinsic}} \sim \exp(-E_{\text{gap}}/2k_B T)$ where I_{OFF} is the minimum current measured inside the bandgap and $n_{\text{intrinsic}}$ is the intrinsic carrier concentration of thermally excited carriers (Kittel, *Introduction to Solid State Physics*. 8th ed., John Wiley & Sons, 2004). In conventional semiconductor physics, the variation of I_{OFF} vs. T can be used to determine E_{gap} . In our case, since the temperature cannot be changed, the variation of I_{OFF} with perpendicular field at constant T can be used to confirm the field dependence of E_{gap} .

To carry out these I_{OFF} vs. E_{gap} measurements, we used a new device, 4L WSe₂, and recorded I_{ds} vs. (V_b, V_t) (Fig. R3a). We found I_{OFF} (minimum current inside the bandgap) at each ΔV . We then plot I_{OFF} vs. bandgap (E_{gap}) corresponding to that ΔV using methods from the main text (Fig. 3). As expected, I_{OFF} drops exponentially when the bandgap increases (Fig. R3b,c). Finally, we fit our data with $I_{\text{OFF}} = I_0 \exp(-E_{\text{gap}}/2k_B T)$ with temperature T as a free parameter (Fig. R3c, red line). From the fit, we obtain $T \approx 440 \pm 10$ K, close to the measurement temperature of ~ 300 K.

Fig. R3 | Bandgap-dependent OFF current. a) I_{ds} (V_g) at several values of ΔV from our 4L device map in a linear scale. **b)** The same plot in log scale. **c)** The off-state current vs. bandgap with an exponential fit (red line).

We find this agreement as good, given that our simple model does not account for effects from contact resistance (which dominate I_{OFF} for small gaps), the contribution of in-gap states (which are large when the gap is almost closed), or leakage through e.g. electrolyte (which is small but can dominate the data for large bandgap). Overall, we conclude that these new measurements suggested by this and second reviewers provide a new, independent verification of field-dependent bandgap closing reported in the main manuscript. We include Fig. R3 together with its discussion in the supplementary information (Fig. S9 and section S6).

Reviewer #2:

The paper by Benjamin Weintrub and colleagues titled “Generating extreme electric fields in 2D materials by dual ionic gating” reports ionic gating of suspended TMDs from two sides. In other words, it substitutes solid dielectrics in traditional dual-gate geometry with electric double-layer transistor dual-gate geometry with much higher capacitance.

I believe that this is an important advancement in the field of ionic liquid gating and that the results (if some of them would be thoroughly justified during the revision process, see below) are general enough and interesting enough to the general audience to grant publication in Nature Communications.

However, before agreeing that this paper is ready for publication, I believe that the following most critical aspect of the work should be addressed:

1) The claim of closing the gap with electric field in trilayer WSe2 reported in Figure 4 should be justified by either (i) temperature-dependent measurements, where the difference in temperature-dependent behavior at different doping levels and electric fields is discussed, or by (ii) optical measurements, where either PL or reflectance should be measured as a function of both gates. Room temperature transport alone cannot serve as a smoking gun evidence of gap closing in multilayers of TMDs. I do understand that temperature-dependent measurement (preferably four-terminal for careful accounting of contact resistance) in dual gate ionic liquid gated sample is extremely challenging. In case authors could perform PL or dR/R measurements instead and demonstrate clearly how optical response changes this would be a much stronger claim. Otherwise, the absence of a resistive region on the bottom of the 2D map in Figure 4 could be explained trivially by the formation of two conductive layers on top and bottom surfaces but without modification of electronic structure. To prove that electronic structure is indeed changing it should be probed in an alternative way (either optics or T dependent transport measurements). The same question is valid for the interpretation of data on 2L WSe2 in Figure 3. If this open question is addressed, the paper will become much stronger.

We followed this suggestion and probed bandgap modification in two alternative complementary ways. First, we carried out a version of the temperature-dependent measurement that was suggested by both reviewers 1 and 2. Since our measurement involves ionic liquids, the temperature cannot be changed directly in a wide range (the ionic liquid either freezes at low temperatures or degrades at high ones). However, we can do a very related measurement. Consider the “off” (subthreshold) current of our device, when the Fermi level is inside the bandgap, E_{gap} . That current is given by $I_{\text{OFF}} \sim n_{\text{intrinsic}} \sim \exp(-E_{\text{gap}}/2k_B T)$

where I_{OFF} is the minimum current measured inside the bandgap and $n_{\text{intrinsic}}$ is the intrinsic carrier concentration of thermally excited carriers (Kittel, *Introduction to Solid State Physics*. 8th ed., John Wiley & Sons, 2004). In conventional semiconductor physics, the variation of I_{OFF} vs. T can be used to determine E_{gap} . In our case, since the temperature cannot be changed, the variation of I_{OFF} with perpendicular field at constant T can be used to confirm the field dependence of E_{gap} .

To carry out these I_{OFF} vs. E_{gap} measurements, we used a new device, 4L WSe₂, and recorded I_{ds} vs. (V_b, V_t) (Fig. R3a). We found I_{OFF} (minimum current inside the bandgap) at each ΔV . We then plot I_{OFF} vs. bandgap (E_{gap}) corresponding to that ΔV using methods from the main text (Fig. 3). As expected, I_{OFF} drops exponentially when the bandgap increases (Fig. R3b,c). Finally, we fit our data with $I_{\text{OFF}} = I_0 \exp(-E_{\text{gap}}/2k_bT)$ with temperature T as a free parameter (Fig. R3c, red line). From the fit, we obtain $T \approx 440 \pm 10$ K, close to the measurement temperature of ~ 300 K.

Fig. R3 | Bandgap-dependent OFF current. a) $I_{\text{ds}}(V_g)$ at several values of ΔV from our 4L device map in a linear scale. **b)** The same plot in log scale. **c)** The off-state current vs. bandgap with an exponential fit (red line).

We find this agreement as good, given that our simple model does not account for effects from contact resistance (which dominate I_{OFF} for small gaps), the contribution of in-gap states (which are large when the gap is almost closed), or leakage through e.g. electrolyte (which is small but can dominate the data for large bandgap). Overall, we conclude that these new measurements suggested by this and second reviewers provide a new, independent verification of field-dependent bandgap closing reported in the main manuscript. We include Fig. R3, along with its detailed discussion, in the supplementary information (Fig. S9 and section S6).

Second, we carried out suggested optical measurements suggested by the reviewer. In general, we agree that optics is the most direct approach to measure bandgaps. There are several complications associated with applying these measurements to our system. First, interlayer excitons in WSe₂ (the energy of which can be used to detect modification of the interlayer bandgap) do not appear in room-temperature photoluminescence (PL) or reflectivity measurements of our samples due to a low interlayer exciton oscillator strength (e.g. *Nano Lett.* 18, 137, 2018). Interlayer excitons in another TMDC bilayer, MoS₂ have higher oscillator strength and can be measured in reflectivity (*Nature Nano.* 16, 888, 2021), but for hBN encapsulated devices at low temperatures. Second, the ionic liquid that we use

luminesces weakly in the range of interlayer excitons. While this PL is very weak, it dominates weak interlayer excitons if they were to appear.

To combat both problems while following the review's suggestion, we performed PL measurements on a related system which is free from both problems. First, instead of bilayer WSe_2 , we fabricated a suspended $\text{MoSe}_2/\text{WSe}_2$ heterobilayer. In that device, an interlayer exciton (which exists across interlayer bandgap and can be used to measure its size) is the lowest energy state, and it has a large enough oscillator strength to be resolved at room temperature in PL measurements. Second, we used NaCl dissolved in water as our electrolyte instead of our usual ionic liquid. This NaCl solution has a much lower capacitance of the ionic liquid/2DM interface, $\sim 0.3 \mu\text{F}/\text{cm}^2$ (*Nano Lett.* 12, 2931, 2012), compared to our DEME-TFSI ionic liquid, $>10 \mu\text{F}/\text{cm}^2$ (*Nat. Rev. Phys.* 3, 508, 2021), and the maximum field from the NaCl solution is correspondingly lower than that from the ionic liquid. While less effective at producing fields than ionic liquids, that system is free from spurious photoluminescence and can be used with an immersion objective, thereby enabling high-resolution measurements. While the details of this system are of course different from that used in the manuscript, optical measurements of the $\text{MoSe}_2/\text{WSe}_2$ heterobilayer can be used to confirm the ability of our ionic gating approach to controllably generate fields.

To probe the field-dependent bandgap of this device, we examined the PL from our $\text{MoSe}_2/\text{WSe}_2$ heterostructure suspended in 10 mM NaCl dissolved in water vs. $\Delta V \equiv V_b - V_t$. To record PL, we used a high numerical aperture objective immersed into the electrolyte. We recorded the data shown below, which we also include in the supporting information (Fig. S10).

Fig. R4 | Field-dependent interlayer exciton photoluminescence of a MoSe₂/WSe₂ heterobilayer suspended in saltwater. a) Interlayer exciton photoluminescence of the suspended heterobilayer as a function of ΔV . The interlayer exciton blueshifts by ~ 15 meV from 0 V to 0.5 V, corresponding to a field of ~ 25 mV/nm, closely matching predictions based on the literature. The feature near 1.26 eV is an artifact, a smoothed-out cosmic ray. **b)** Cartoon depicting orientation of the voltage relative to the TMDC layers. A positive voltage will cause the interlayer bandgap to increase, thereby causing the excitons to blueshift.

At zero ΔV , we observe a peak at ~ 1.335 eV, which is ascribed to interlayer excitons, in addition to other features at higher energies corresponding to in-plane excitons. With increased ΔV up to 0.5 V, the peak blueshifts by $\Delta E = 15$ meV to ~ 1.350 eV at the edge of the electrochemical window for this electrolyte ($|\Delta V| < 0.5$ V). This is the expected result of increased interlayer field oriented as shown in the above cartoon from MoSe₂ to WSe₂. We note that the symmetry between the layers of the MoSe₂/WSe₂ heterobilayer is broken even at zero field, unlike in homobilayer WSe₂. The application of a field in one direction redshifts the interlayer exciton, and a field in the other direction blueshifts it (*Science* 366, 870, 2019). As the oscillator strength is higher for the blueshifted interlayer exciton, it is easier for us to observe it. We estimate the field inside the heterostructure from interlayer exciton position modification as $F_{\perp} = \Delta E / e d_{\text{int}} = 25$ mV/nm, where $d_{\text{int}} \approx 0.6$ nm is the interlayer distance.

We can also compare the field measured in this way to the expected field inside the ionic double layer, which is estimated using the same techniques as in the manuscript. We can estimate the thickness of the ionic double layer, the Debye length, at our molar concentration as ~ 7.5 nm using previous work (*Nano Lett.* 12, 6, 2012). We then estimate a maximum field strength of $F_{\perp} \approx \Delta V / (2d_{\text{Debye}}) \approx 30$ mV/nm, very close to the value obtained from the shift of the interlayer exciton energy.

In summary, these optical measurements suggested by the reviewer show that a simple water-based electrolyte used here obviously cannot compete in field strength with the ionic liquid, DEME-TFSI, used in the manuscript. Nevertheless, optical measurements provide another independent experimental confirmation that double-sided ionic gating produces an out-of-plane electric field and that the strength of this field conforms to the model for field strength used in the manuscript.

The reviewer also says “the absence of a resistive region on the bottom of the 2D map in Figure 4 could be explained trivially by the formation of two conductive layers on top and bottom surfaces but without modification of electronic structure.”

Here, we wanted to clarify for a true tunable-bandgap bilayer system under perpendicular field, the wavefunctions associated with conduction and valence bands are localized in the opposite layers of the bilayer system. One of the most direct experimental confirmations of this is the out-of-plane orientation of the dipole moment of interlayer excitons in bilayer systems (*Sci. Rep.* 12, 6939, 2022). Because of that, using the classical language one can say that once the Fermi level is in the conduction (valence) band of a bilayer in a field, a conductive

region in the top (bottom) layer of the bilayer arises. But in the quantum language, the bilayer in a field behaves as compound material with a field-tunable bandgap.

To better visualize this point, we developed a new approach (based on model from *Phys. Rev. Lett.* 123, 117702, 2019) to calculate the *layer-resolved* carrier density of multilayer TMDCs in the presence of electric fields. The simulation solves for carrier densities inside each layer of the multilayer TMDC so that the total chemical potential is equilibrated throughout the device (details in the updated supplementary material). The field shifts the band structure of one layer relative to the adjacent layer by $\Delta E = ed_{\text{int}}F_{\perp}$. The model is widely used in the literature and produces data consistent with DFT calculations (e.g. *Phys. Rev. B* 84, 205325, 2011 and *Nano Lett.* 18, 137, 2018).

First, we checked that this new model produces $I_{\text{ds}}(V_t, V_b)$ that is very similar to a more simple model used in the previous version of the manuscript (please see our answer to the first reviewer for more details). Second, we can now explicitly resolve how the carrier density is distributed inside the bilayer for different fields (Fig. R5 below). We map the carrier density in the bottom layer, the top layer, and the sum of the absolute value of bottom/top carrier densities (which determines the current detected a transport experiment), all plotted vs. (V_t, V_b) . The simulations show that the carrier density of the top or bottom layer becomes non-zero in regions adjacent to the bandgap region. The simulation also confirms that the interlayer bandgap of the homobilayer is field-dependent. We include this information in the supplementary information (Fig. S4).

Fig. R5 | Layer-resolved carrier density in 2L WSe₂. The simulated carrier density in **a)** the bottom and **b)** top layers of bilayer WSe₂. **c)** Total carriers for the system. The diamond in each figure corresponds to the Fermi level inside the bandgap of the entire system.

This is how any semiconducting homobilayer system under strong fields should work: a strong-enough field (meaning large enough $|\Delta V|$) “moves” all available carriers into their respective layer, if these carriers are present.

2) Related to previous point, but on a more abstract note, which will echo concerns of other reviewers – authors do say in the introduction to the paper that ultra-high electric fields could enable a lot of new experiments, but they do not show enough evidence for that. In case my point 1) is addressed then there would be more substantial evidence that it is indeed

possible to do something exciting with those electric fields (change substantially or close the bandgap of TMDs).

Regarding the introduction, we wanted to emphasize that multiple theoretical works in high profile journals require such fields. It is our intent to work on these possibilities as our next project. While the paper was in review, new possibilities for high field applications were pointed out. For example, in (*Phys. Rev. B* 105, 174411, 2022) 1L and 2L Fe is predicted to have a field-dependent magnon dispersion, while in (*Nat. Phys.* 18, 395, 2022) a strong perpendicular field would enable the same type of experiment without the need for an ultra-thin hBN spacer layer thereby enabling measurements with higher oscillator strength. Our approach provides strong enough fields or paves the way to access them. We also wanted to point out again that we do show one example of new physics at strong fields, namely bandgap closing of TMDs (Figs. 3-4 of the main text). To reflect this view, we added references to these new possibilities in lines 63 to 65 of the main text.

Some other more technical questions:

3) To understand the methodology of reference potential measurement, have the authors tried to use a high impedance voltmeter (for example electrometer) instead of sourcing additional voltage to a reference electrode (which could act as a gate as well)? It might be worth providing a bit more details regarding the measurement of potential inside IL for general readers. It might be also worth explaining if biasing reference electrodes affects the distribution of potentials inside the IL.

We agree that if a potential is applied to any piece of metal in contact with the IL, that metal should act as a gate. In our measurement configuration, however, the Keithley source meter maintains zero current flow (down to pA) between the liquid and the reference electrodes, and hence their potential is the same. An additional electrometer would increase the accuracy of reference voltage measurement, but the resolution higher than mV is not needed in our experiments. This protocol that we followed became standard in the field and is used in many measurements employing ionic liquids. We now add the extra reference, (*Nat. Rev. Phys.* 3, 508, 2021), to the supplementary materials. As an extra precaution, we make the area of the bottom reference electrode very small compared to the area of the bottom gate electrode, thereby reducing its influence on gating the bottom IL.

To further show that our reference voltage data is robust, below in Fig. R6 we show a) the field extracted from the transport map and b) the field extracted from reference voltage measurements for our new 4L device. The two measurements yield very similar results in the region where they can be compared. This indicates that any potential influence on reference voltage measurements from the measurement device is small. We include this new data in the supplementary information (Fig. S11), since we feel it helps validate our approach.

Fig. R6 | Field from transport vs. field from reference voltage in 4L WSe₂. The field extracted from **a)** transport measurements is very similar to the field extracted from **b)** reference voltage measurements, they both show fields corresponding to bandgap closing (~ 0.9 V/nm) at $\Delta V \approx 4$ V. This shows that both measurement approaches can accurately measure the field through a (charge-neutral) device.

4) SI Figure 3 contains a 2D map and linecuts for dual ionic liquid gating in monolayer WSe₂. What does the decrease of current to zero at negative gate voltages represent in linecuts? Is that degradation?

These data points correspond to the first few lines taken of the entire map, and if the device has not been used in a while, it needs a few scans to “wake up”. This likely corresponds to adsorbates and other contaminants located on the surface of the 2DM and/or gate electrodes, which tends to go away with use. This behavior is typical for measurements with ionic liquids. We do not believe it is degradation, especially since the response gets stronger with subsequent lines of the map. To make this clearer to the reader, we added the following sentence to the figure description. “The first few scans of the map in **a)** around $V_b \approx -3$ V are likely hysteretic features, and they appear in the line cuts in **b)** at the lowest V_g values for each curve. These features should be regarded as outliers to the data”.

5) Have authors measured the capacitance of their ionic liquids?

To address this comment, we carried out additional photoluminescence measurements of single-gated devices in our ionic liquid (Fig. R7). We tracked the trion peak, the position of which is known to depend on carrier density. Specifically, we followed the procedure described in *Phys. Rev. Lett.* 115, 126802, 2015 to extract the carrier density (n) from an applied gate voltage (V_g). The n vs. V_g data obtained this way is shown in the figure below. From the linear fit to that data, we calculate the areal capacitance of the ionic liquid/2DM interface to be ~ 10 $\mu\text{F}/\text{cm}^2$. This value is similar to literature values (*Nat. Rev. Phys.* 3, 508, 2021). We include this data in the supplementary information (Fig. S12).

Fig. R7 | Extracting areal gate capacitance. **a)** The trion peak redshifts as more charges are brought into the material, allowing us to obtain **b)** carrier density vs. gate voltage data. From the carrier density vs. gate voltage data, we obtain a gate areal capacitance of $\sim 10 \mu\text{F}/\text{cm}^2$.

Some technical comments to better represent the data:

6) Figures 1 and 2 could be merged. It is up to the authors of course, but they discuss two very related things in the form of schematics, and it could be beneficial for a potential reader to see the concept, physics of what is happening, and device schematic and image on the same figure. At the same time, each figure taken individually does not contain enough information in my opinion.

We thought hard about this and made sketches of possible figures. While we fully see the reviewer's perspective, our feeling is that such a figure would deviate from the discussion in the text. Fig. 1 on our view shows the concept of the device, and Fig. 2 shows device implementation. Since the implementation is complex and includes many parts, we did not want to give the reader a false impression that the idea behind that device is equally complex. And we felt that the two figures represent two separate informational units that should be kept distinct.

7) Supplementary Figure 1 – it might be worth increasing the size of device images, they are not readable, also please add scale bars.

We agree. We enlarge the pictures and added scalebars. We decide to only show one device, as other devices look similar. The updated Fig. S1 is shown below, Fig. R8, for convenience.

Fig. R8 | Sample #1, 2L WSe₂, before and after measurement **a)** The bilayer device before measurement. The black square is the suspended region with side length $\sim 4 \mu\text{m}$ (length of the scale bar). **b)** Tilted SEM image of the device after measurement (scale bar is $1 \mu\text{m}$). The image indicates that the device is not ruptured, and the PMMA visibly covers the metal and supported regions of the flake.

8) Grey lines on 2D maps in Figure 3 and 4 should be described in the main text or Figure captions.

We have removed the gray lines in the maps to improve clarity (Figs. 3, 4, and S3).

Reviewer #3:

The points raised in the previous round of review have been satisfactorily addressed.

A significant advance of the present work over previous studies of tunable bandgaps in 2D materials using vertical electric fields -such as Nature 459, 820–823 (2009), Nano Lett. 15, 8000–8007 (2015), is that a much stronger electric field is achievable using dual ionic-liquid gates in this work.

However, I feel that the manuscript can be further improved by incorporating data the authors might have already taken.

1) As the bandgap is reduced by a vertical electric field, the off current is expected to increase. The authors should also replot Figure 3b on logarithmic scale to show how the off current varies with the vertical electric field in the supplementary information. Similar plots should also be made for the trilayer device.

We fully agree. Since the off current for our 2L device is below the noise floor of our measurement equipment and our 3L device data has too low resolution to accurately resolve this effect, we recorded the off current as a function of electric field in a new 4L WSe₂ device and found that it increases as suggested. In fact, we now use the analysis of the off current near bandgap closing to independently confirm the validity of our bandgap measurements. A more detailed analysis is in the answer to the fourth point of the first reviewer and the first point of the second reviewer.

2) Did the authors measure the authors measure the PL of their bilayer devices as a function of ionic-liquid gate voltage? If so, did they observe the redshift of a peak corresponding to the interlayer transition, like figure 4 in Nano Lett. 15, 8000–8007 (2015)?

Following the suggestion as well as the question of the second reviewer, we carried out field-dependent photoluminescence measurements. Very briefly, the experimental limitations forced us to use a different electrolyte system and a different interlayer exciton. Nevertheless, we now do observe the expected shift of the interlayer exciton energy with applied field. The magnitude of the shift is in agreement with the expectation from double-sided ionic gating. The details of this measurement are discussed in our answer to point 1) part ii) of the second reviewer.

REVIEWERS' COMMENTS

Reviewer #1 (Remarks to the Author):

After reading through the reply and modification, I believe this paper has reached acceptance status for the present revision. This doesn't mean that everything is convincing. Especially the answer to my question about Hall measurement is far from getting satisfaction. I understand this as still a healthy part of scientific exploration. The claim is still exciting and inspiring to the whole community. I believe It's time to publish the manuscript and let the community discuss the results.

Reviewer #2 (Remarks to the Author):

I believe that the authors fully addressed my comments and concerns. I am especially impressed that additional optical measurements were performed by the authors and that they did provide independent evidence of bandgap modification in TMDs. I now recommend this paper for publication in Nature Communications.

Reviewer #3 (Remarks to the Author):

The authors have adequately addressed the questions/comments raised in the previous round of review. Particularly, they used two independent approaches (optical and transport measurements) to confirm modification and closing of the bandgap in response to a perpendicular field. They also included screening by free carriers in their model to extract the bandgap and field strength from transport data.

This version is significantly improved over the previous version.